# Mitoregulin Promotes Cell Cycle Progression in Non-Small Cell Lung Cancer Cells

**DOI:** 10.3390/ijms26051939

**Published:** 2025-02-24

**Authors:** Colleen S. Stein, Connor R. Linzer, Collin D. Heer, Nathan H. Witmer, Jesse D. Cochran, Douglas R. Spitz, Ryan L. Boudreau

**Affiliations:** 1Department of Internal Medicine, Carver College of Medicine, University of Iowa, Iowa City, IA 52242, USA; Clinzer@mcw.edu (C.R.L.); nathan-witmer@uiowa.edu (N.H.W.); jc6gdt@virginia.edu (J.D.C.); 2Free Radical and Radiation Biology Program, Department of Radiation Oncology, Holden Comprehensive Cancer Center, University of Iowa, Iowa City, IA 52242, USA; collin.heer@yale.edu (C.D.H.); douglas-spitz@uiowa.edu (D.R.S.)

**Keywords:** mitochondria, long non-coding RNA, oxidative stress, micropeptide, oxidant, prooxidant, miR-17, cancer therapy, cell cycle arrest, *LINC00116*

## Abstract

Mitoregulin (MTLN) is a 56-amino-acid mitochondrial microprotein known to modulate mitochondrial energetics. MTLN gene expression is elevated broadly across most cancers and has been proposed as a prognostic biomarker for non-small cell lung cancer (NSCLC). In addition, lower MTLN expression in lung adenocarcinoma (LUAD) correlates with significantly improved patient survival. In our studies, we have found that MTLN silencing in A549 NSCLC cells slowed proliferation and, in accordance with this, we observed the following: (1) increased proportion of cells in the G1 phase of cell cycle; (2) protein changes consistent with G1 arrest (e.g., reduced levels and/or reduced phosphorylation of ERK, MYC, CDK2, and RB, and elevated p27^Kip1^); (3) reduction in clonogenic cell survival and; (4) lower steady-state cytosolic and mitochondrial H_2_O_2_ levels as indicated by use of the roGFP2-Orp1 redox sensor. Conflicting with G1 arrest, we observed a boost in cyclin D1 abundance. We also tested MTLN silencing in combination with buthionine sulfoximine (BSO) and auranofin (AF), drugs that inhibit GSH synthesis and thioredoxin reductase, respectively, to elevate the reactive oxygen species (ROS) amount to a toxic range. Interestingly, clonogenic survival after drug treatment was greater for MTLN-silenced cultures versus the control cultures. Lower H_2_O_2_ output and reduced vulnerability to ROS damage due to G1 status may have jointly contributed to the partial BSO + AF resistance. Overall, our results provide evidence that MTLN fosters H_2_O_2_ signaling to propel G1/S transition and suggest MTLN silencing as a therapeutic strategy to limit NSCLC growth.

## 1. Introduction

Cancer continues to be the second leading cause of death within the US, with estimations of ~2 million new cases and ~600,000 cancer-related deaths in 2024 [1]. Lung and bronchus cancer account for the largest percentage of cancer-related deaths, at roughly 21% [1]. Non-small cell lung cancer (NSCLC) is the primary type of lung cancer at 80%, with lung adenocarcinoma (LUAD) being the most prominent NSCLC subtype [2]. Despite a decline in incidence of and deaths from lung cancer in recent years, attributable in part to the continuing decrease in cigarette smoking, lung cancer remains one of the deadliest cancers with a 1-year survival rate at less than 50% [2]. There remains a dire need to identify novel players and targetable pathways that can drive lung cancer cell proliferation and cancer progression.

An emergent avenue for investigation is the study of newly discovered microproteins [3]. Studies have found that DNA gives rise to thousands of uncharacterized RNAs, including long non-coding RNAs (lncRNAs), which are >200 nucleotides and thought to lack protein-coding open-reading frames (ORFs). However, updated analyses have revealed that many lncRNAs have been misannotated and can, in fact, code for microproteins (<100 amino acids, below the original ORF cutoff) [4]. Despite their recent discovery, we and other groups have already identified fundamental cellular roles for microproteins across diverse processes and disease states [5,6,7]. With the possibility of uncovered microprotein functions relating to cell metabolism and oxidative stress, microproteins present an attractive area for cancer research.

Chen et al. recently utilized expression profiling to describe the “non-coding” RNA *LINC00116* as a novel prognostic biomarker of NSCLC [8]. We, along with five other groups, discovered that *LINC00116* was misannotated as “non-coding” and contains a highly conserved (fish-to-humans) ORF that codes for a 56-amino-acid microprotein that localizes to mitochondria [7,9,10,11,12,13] and has officially been named mitoregulin (MTLN, protein; *MTLN*, gene). Although MTLN’s function at the molecular level is not fully understood, there is evidence that MTLN impacts broad mitochondrial processes, including oxidative phosphorylation (OXPHOS), reactive oxygen species (ROS) generation, calcium handling, and lipid metabolism [7,9,10,11,12,14,15,16,17,18,19].

With *LINC00116*’s description as a biomarker for NSCLC, we were inspired to investigate the relationship between MTLN and NSCLC cell proliferation. Proliferative signals begin with a growth factor or mitogen binding to receptors at the plasma membrane and signal propagation via the RAS-RAF-MEK-ERK pathway. Phosphorylated (activated) ERK (p-ERK) is translocated to the nucleus to induce the expression of genes for cell cycle entry, such as G1 cyclins and cyclin-dependent kinases (CDKs) [20]. During G1 progression, the cyclin D1/CDK4/6 complex and the cyclin E/CDK2 complex phosphorylate nuclear retinoblastoma protein (RB), triggering its release from the E2F transcription factor, for the de-repression of E2F-driven genes that prepare cells for DNA replication and division. Cyclin E/CDK2 activity peaks late in G1 and is essential as an amplification loop to trigger a “no-return” entry into S phase [21,22]. Also critical to cycle regulation are CDK inhibitors, p27^Kip1^ and p21, which bind cyclin/CDK complexes to block kinase activity [21,22]. A sharp drop in p27^Kip1^ via the ubiquitin–proteasome system (UPS) coincides with a rise in cyclin/CDK2 activity to drive a switch-like entry into the S phase [23,24]. Another key player is the oncogenic transcription factor c-myc (MYC) [25,26,27]. MYC is induced by growth factor stimulation and functions in a positive-feedback fashion, with p-ERK1/2 to promote cycle entry. Cancer cells acquire mutations that drive uncontrolled growth. The NSCLC/LUAD A549 cell line harbors the c.34G>A/p.G12S KRAS mutation [28], which is one of the most frequent oncogenic mutations across cancers and particularly prevalent in LUAD [29]. This mutation imparts a sustained activation status to KRAS to prolong the (K)RAS-RAF-MEK-ERK phosphorylation-based signaling cascade and perpetuate cell cycle progression.

An additional key player that cooperates along proliferative signaling pathways is H_2_O_2_. Low level H_2_O_2_ is known to stimulate ERK activation to help drive cell cycle entry and G1/S transition [30,31]. Mitochondria are a major source of intracellular ROS, liberating low concentrations as a normal by-product of OXPHOS [32], and mitochondrial-derived ROS has been shown to be critical for cell cycle progression [31,33,34,35].

In this current study, we queried public databases and found that MTLN gene expression is elevated in most cancerous tissues including LUAD. We examined the proliferation and the cell cycle status in the A549 LUAD cell line, which has abundant MTLN expression, and determined that MTLN knockdown (KD) induces G1 phase cell cycle arrest in correlation with lowered basal H_2_O_2_. We also explored the effects of treating A549 cells with buthionine sulfoximine (BSO) and auranofin (AF), which act by blocking the GSH and thioredoxin antioxidant pathways, respectively [36,37,38]. These drugs severely diminished A549 colony forming capacity (CFC), and MTLN KD partially reversed this effect. Together, these findings support the hypothesis that MTLN KD diminishes growth-stimulatory H_2_O_2_ production, thereby reducing A549 proliferative capacity and providing some resistance to ROS-elevating drugs.

## 2. Results

### 2.1. MTLN Expression Is Increased in Most Cancer Types and Inversely Correlates with Overall Survival in LUAD

Given the observed influence of MTLN in the key processes of cancer cells (ROS modulation, energy production, overall mitochondrial function), we began by examining the expression of MTLN across cancer types. Using TNMplot, a web tool that plots mRNA expression across cancerous and non-cancerous tissues [39], it was evident that gene expression of MTLN was significantly higher in cancerous tissue compared to the neighboring non-cancerous tissue for most cancer types (14 of 22) (Figure 1A). We focused our investigation on lung cancer, and specifically on LUAD, for which MTLN gene expression is significantly increased compared to normal lung tissue (Figure 1B; data acquired from the OncoDB website [40]). With the use of an online tool [41,42] we generated a Kaplan–Meier plot for LUAD, which strikingly shows that low-level MTLN gene expression is a strong predictor of improved overall survival (Figure 1C). These MTLN findings are consistent with and reinforce a previous report examining *LINC00116* (*MTLN*) expression in a distinct cohort of 19 paired lung cancer and non-cancerous adjacent tissue [8].

### 2.2. MTLN Knockdown Induces G1 Phase Cell Cycle Arrest in A549 Cells

With an interest in understanding how MTLN, as a mitochondrial microprotein, influences tumor cell survival in LUAD, we examined the effects of manipulating MTLN expression in the A549 NSCLC/LUAD cell line. Using a custom antibody that recognizes the c-terminal region of MTLN [7], we first confirmed the expression of endogenous MTLN in A549 cells by immunostaining. MTLN was readily detected and co-localized with mitochondria labeled with MitoTracker^TM^ (Figure 2A). The KD of MTLN using siRNA oligos (siMTLN) showed a clear loss of MTLN staining (Figure 2A), confirming specificity of the antibody. For co-localization, we determined a Pearson’s correlation coefficient of 0.8, confirming a strong overlap of MTLN and mitochondria fluorescence intensities. This coincides with a multitude of previous reports from us and others demonstrating MTLN residence in mitochondria in normal and/or cancer cells [7,10,11,12,13,14,18,19]. These results indicate that *MTLN*/*LINC00116* transcripts are actively translated within LUAD cells and implicate MTLN microprotein, as opposed to a non-coding transcript, in modulating LUAD cancer outcomes. Of course, future efforts are needed to rigorously verify that increased MTLN mRNA levels (Figure 1A,B) are also reflected at the protein level in cancerous tissue samples relative to non-cancerous neighboring tissue samples.

We next addressed MTLN influence on LUAD cell proliferation. We acutely silenced MTLN in exponentially growing A549 cells and by 2 days post-KD (3 days post-seeding), the cell counts diverged, with significantly lower numbers recovered after the application of siMTLN compared to the negative control RNA oligos (siCont) (Figure 2B). Flow cytometric analyses of the DNA content after propidium iodide staining at 48 h post-KD revealed increased G1 and reduced S and G2 phase populations in siMTLN compared to siCont or mock treatment (addition of transfection reagent in the absence of siRNA oligos) (Figure 2C). The mean G1 proportions were 56.6 ± 2.0%, 76.1 ± 1.0%, and 49.5 ± 1.5% for the siCont, siMTLN, and mock treatment groups, respectively (Figure 2C), indicative of G1 cycle arrest and consistent with the blunted proliferation rate. Together, these results indicate a role for MTLN in promoting G1-to-S cell cycle progression.

To gain insight into the mechanism of G1 arrest via MTLN KD, we performed immunoblot analyses for the proteins involved in cell cycle regulation (Figure 2D). In accordance with a G1 arrest, we detected significantly reduced levels of p-ERK (T202/Y204 phospho-p44/42 MAPK ERK1/2), cyclin E, CDK2, phosphorylated RB (p-RB; phospho-S807/811), and MYC, all key players in driving G1 to S transition. In addition, the p27^Kip1^ level was significantly increased, consistent with its role as a CDK inhibitor and diminished p-RB. The level of p21, another CDK inhibitor in the KIP family, was unaffected by MTLN KD (Figure 2D). Unlike p27^Kip1^, p21 is a p53 target gene, induced to stall cell cycle after DNA damage [43]. For A549 cells, a blockade of ERK phosphorylation has been known to trigger compensatory AKT activation due to KRAS crosstalk with the PI3K-AKT pathway [44,45]. After MTLN KD, however, AKT activation, as indicated by T308 phosphorylation [46], was not elevated, but rather was slightly repressed (Figure 2D). This suggests that MTLN’s influence may extend beyond ERK pathway inhibition to mildly down-modulate the AKT pathway, further antagonizing growth. Interestingly, and counterintuitive to G1 arrest, cyclin D1 levels were significantly increased upon MTLN KD (Figure 2D). Presumably, p27^Kip1^ was sufficiently elevated to block the kinase activities of the cyclin D1-CDK4/6 and cyclin E-CDK2 complexes, as evidenced by the diminished p-RB. When we assayed another NSCLC cell line, H1299, for changes in the key cell cycle-related proteins after MTLN KD, we observed similar decreases in p-RB and MYC and an increase in p27^Kip1^, indicative of G1 arrest, as well as an increase in cyclin D1 (Appendix A). Collectively, these data show that the loss of MTLN arrests NSCLC cells in G1 by altering the abundance or phosphorylation status of critical regulators of G1-to-S progression.

### 2.3. MTLN Knockdown Decreases Steady-State Cytosolic and Mitochondrial Oxidation of roGFP2-Orp1, Indicative of Decreased H_2_O_2_ Levels in A549 Cells

ROS such as H_2_O_2_, have been described as second messengers and regulators in normal and cancer cell proliferation [32,47,48,49,50]. Intracellular ROS levels are tightly regulated and produce distinct effects in a dose-dependent fashion; low level ROS mediate physiological responses, while high levels can form toxic radicals to cause cell stasis or irreversible damage that triggers cell death [51]. With a mitogenic stimulus, H_2_O_2_ levels are moderately elevated to promote cell proliferation, and blocking H_2_O_2_ prevents G1/S phase transition [47,52,53]. To gain real-time information on relative cellular H_2_O_2_ concentrations after MTLN KD, we used roGFP2-Orp1, a ratiometric H_2_O_2_ biosensor [54]. roGFPs have an engineered dithiol/disulfide switch that, when oxidized to disulfide formation, shifts the optimal excitation wavelength from 488 nm to 400 nm [55]. The roGFP2-Orp1 sensor consists of roGFP2 translationally fused to the yeast thiol peroxidase Orp1, which efficiently reacts with H_2_O_2_ and transmits oxidation to roGFP2 by a thiol–disulfide exchange reaction [56]. Fluorescence intensity readings (at the common 520 nm emission wavelength) are recorded for both 400 nm and 488 nm excitation wavelengths, and the ratio of these readings (oxidized/reduced) reflects H_2_O_2_ levels. We used adenovirus vectors to introduce either cytosolic- or mitochondrial-targeted roGFP2-Orp1 [54] to A549 cells along with siMTLN or siCont treatment. Fluorescence measurements taken at 48 h post-KD revealed a significantly lower oxidized/reduced ratio for MTLN KD compared to siCont in both the cytosolic and mitochondrial compartments (Figure 3A). This indicates that mitochondrial H_2_O_2_ output is diminished by MTLN silencing and may drop below a threshold necessary to drive G1 to S transition. We attempted to restore proliferation rate by spiking A549 cultures with H_2_O_2_; MTLN KD or control cells were exposed to varying H_2_O_2_ doses for 3 h then harvested and counted 24 h later. However, this was ineffective (Figure 3B) and supports the likelihood that bolus dosing of H_2_O_2_ cannot reproduce the physiologic situation of continuous metabolic release of H_2_O_2_ when it comes to driving cell cycle progression. It is also possible that diminished H_2_O_2_ is not the only factor contributing to slowed cell cycle in MTLN-silenced cells. As expected, H_2_O_2_ became toxic with an increased dose (Figure 3B). Interestingly, significant growth inhibition began at 60 μM for siCont but not until 120 μM for siMTLN cultures. Moreover, siCont cell counts dropped more steeply with progression to 480 μM (54% vs. 48% cell loss relative to “no H_2_O_2_” for siCont vs. siMTLN), suggesting that MTLN silencing provides some resistance to the harmful effects of higher H_2_O_2_ doses.

Another mitochondrial-localized protein, ROS Modulator 1 (ROMO1), has been shown to be essential to cell cycle progression in cancer- [34,57] and non-cancer [35] cells via the generation of ROS from mitochondrial complex III [58]. ROMO1 KD triggers G1 arrest in several cancer cell lines including H1299 and A549 cells [34]. Conceivably, MTLN and ROMO1 could work in concert within the mitochondria to promote physiological ROS to support proliferation. Consistent with this possibility, ROMO1 levels were moderately but significantly decreased when we silenced MTLN (Figure 2D). Like MTLN, ROMO1 gene expression is elevated in many cancers [59,60]. Pairwise gene expression analysis performed using the OncoDB website tool, demonstrates a high correlation of MTLN and ROMO1 gene expression in LUAD (Appendix A), suggestive of a functional relationship. In a separate experiment, we compared side by side the effects of ROMO1 KD versus MTLN KD in A549 cells (Appendix A). Though the degree of KD was not particularly robust in this experiment, we again observed MTLN KD to prompt rises in cyclin D1 and p27^Kip1^ protein levels. Notably, MTLN KD in this experiment did not elicit a significant drop in ROMO1, indicating that the siMTLN-induced elevation in p27^Kip1^ is not reliant on ROMO1 decline. Contrary to the previous works [34,35], ROMO1 KD in our hands did not lead to a significant drop in cyclin D1 or rise in p27^Kip1^. We suspect stronger KD may be required to generate these reported ROMO1-silencing effects. Together, our MTLN KD data, along with published ROMO1 data [34], imply a dual need for ROMO1 and MTLN to generate mitochondrial ROS for G1/S transition; however, additional interrogation is needed to determine if they operate in parallel or along a common molecular pathway. Intriguingly, our findings reveal a distinct effect of MTLN KD on cyclin D1.

### 2.4. MTLN Knockdown Reduces Efficacy of BSO + AF Drug Treatment

To further investigate the involvement of MTLN in hydroperoxide metabolism, we tested the effect of BSO + AF anti-cancer drugs in the context of MTLN silencing. The combination treatment of BSO + AF is utilized to exploit the high metabolism and robust oxidant production of cancer cells to trigger their own demise [36,37,38]. These drugs block the two major antioxidant pathways in cells. BSO, a GSH depleting agent, and AF, a thioredoxin reductase inhibitor, are utilized to simultaneously inhibit glutathione peroxidase- and peroxiredoxin-mediated H_2_O_2_ scavenging [36,37,38]. We transfected A549 cells with siMTLN or siCont oligos and then added BSO after 48 h; the following day, we exposed the cells to 2.5 μM AF for 3 h. Notably, a small but significant toxic effect was detectable at the 3 h AF exposure timepoint as indicated by lactate dehydrogenase (LDH) leak, for both control- and MTLN-KD cultures, (Figure 4A). LDH release was prevented by N-acetyl cysteine (NAC) antioxidant, verifying the central role of thiol-mediated oxidative stress in the toxicity of the BSO + AF drug combination. For clonogenic assays, the cells were treated as described above then collected after the 3 h AF exposure, re-seeded at low density, and monitored over 2 weeks for colony growth. In the absence of drugs, the plating efficiency (proportion of colony-forming cells relative to number seeded) was substantially reduced by MTLN silencing (Figure 4B). This is in alignment with impaired cell cycle progression (Figure 2B–D). Interestingly, MTLN KD imparted a significant degree of resistance against the BSO + AF treatment, as indicated by a higher surviving fraction (normalized to no-drug plating efficiency) for siMTLN compared to siCont (Figure 4C). This outcome suggests that cell cycle inhibition may lessen the toxic impact of these drugs. Indeed, cells that are arrested in G0/G1, and not actively replicating DNA, are less vulnerable to oxidant-induced DNA damage and maintain higher clonogenicity compared to proliferating counterparts [61]. Moreover, H_2_O_2_ exposure has been shown to create more DNA double-strand breaks and killing in S phase compared to G1 phase cells [62].

To assess DNA damage, cell lysates collected after BSO + AF treatment were blotted for S139-phosphorylated histone H2AX (p-S139 H2AX), a marker of DNA damage. Phosho-S139 H2AX band intensities were strongly augmented after BSO + AF treatment, and no differences in p-S139 H2AX/total H2AX ratios were noted between siCont and MTLN KD (Figure 4D). This result indicates substantial BSO + AF-induced DNA damage for both groups, matching the high proportion of cell loss. This assay likely lacks sensitivity to distinguish subtle differences or a minor population of less-damaged cells that might allow for growth versus no growth/death in the clonogenic assay. Another possibility is that MTLN KD could influence DNA repair mechanisms. A non-canonical role described for cyclin D1 is to recruit DNA repair proteins after radiation-induced damage [63]. The elevated cyclin D1 upon MTLN KD (Figure 2D and Appendix A), while incongruent with G1 arrest, could contribute to the improved colony formation through enhanced DNA repair after drug exposure. A more in-depth determination of DNA damage and repair under ROS-inducing chemotherapeutics in the context of MTLN and cyclin D1 expression level presents an important future direction for follow-up investigations.

We also considered that MTLN KD might blunt the ROS surge triggered by AF to lessen cell death. The roGFP2-Orp1 H_2_O_2_ sensor could not be utilized to address this because AF has been reported to react directly with roGFP2 to form an S-Au-S bridge to mimic disulfide formation [64]. Instead, we used the fluorogenic probe CellROX Orange to assess the ROS levels in live cells by fluorescence microcopy. The control and MTLN KD cultures were treated with 24 h BSO (100 μM) + 3 h AF (2.5 or 5 μM) or with no drugs, and CellROX loading was performed for the final 30 min of the treatment schedule. Without drug treatment, CellROX signal was below the level of detection (Appendix A). With 2.5 μM AF, CellROX fluorescence was detected sporadically in the cells across the siCont wells, while the siMTLN wells contained a significantly higher proportion of positive cells (Appendix A), resulting in a higher mean fluorescence intensity for MTLN-silenced compared to siCont wells (Appendix A). With 5.0 μM AF, most cells in both the siCont and siMTLN wells displayed bright fluorescence (Appendix A), with mean intensities significantly elevated above the untreated control wells (Appendix A). Though not confirmed, the fluorescence signal appeared to be concentrated at the mitochondria, a pattern consistent with the AF-mediated inhibition of mitochondrial thioredoxin reductase which caused increases in the steady-state levels of H_2_O_2_. An interesting observation at 5 μM AF was the occurrence of a CellROX negative (or very dim) population of cells (Appendix A), which was more abundant in siMTLN compared to the siCont wells (Appendix A). These could represent metabolically inactive cells, perhaps coinciding with the G1 arrested population in MTLN-silenced cultures. CellROX Orange is a broad-spectrum ROS detector, with undefined relative sensitivities for various ROS species, limiting the interpretation of these data. In summary, the partial resistance to BSO + AF with MTLN KD likely reflects a subpopulation of cells that, at the time of AF treatment, have low mitochondrial metabolism, such as G1-arrested cells. Furthermore, enhanced cyclin D1 levels could promote DNA repair after ROS damage to salvage additional cells within the MTLN KD cultures.

## 3. Discussion

Through this investigation, we examined MTLN’s relationship with lung cancer, with a focus on the MTLN effects on A549 LUAD cell proliferation and sensitivity to inhibitors of hydroperoxide metabolism. MTLN gene expression is elevated in most cancers, and for LUAD, we found that lower MTLN gene expression correlates with improved overall survival and prognosis. While the data clearly show that the MTLN transcript level is elevated in LUAD vs. normal tissue and is predictive of poor outcome, our readouts are limited in that they do not address whether MTLN gene expression is biased towards any subgroups such as LUAD stage, treatment protocols, or driver mutations. As cancer datasets and data mining capabilities expand, acquiring such information could prove vital to understanding molecular mechanisms and tailoring of any MTLN-based therapeutics.

In examining the effect of MTLN KD on A549 cell proliferation, we determined that KD led to G1 arrest and slowed A549 growth. Consistent with this, substantially reduced levels of p-ERK1/2, p-RB, MYC, cyclin E, and CDK2, and a heightened level of p27^Kip1^ were detected by Western blotting. Figure 5 is a schematic illustrating key points at which MTLN might affect signaling cascades to promote G1/S transition.

Though our study highlights a key role for MTLN in driving the proliferation of lung cancer cells, it does not exclude the possibility that MTLN is recruited to push cell cycle progression in other pathologies. For example, MTLN is upregulated in chronic kidney disease and promotes fibrosis [65]. That study does not specifically address proliferation, but it is conceivable that MTLN plays a role in fibroblast proliferation. Involvement of MTLN in promoting cell cycle progression in diseases apart from cancer is an intriguing area open to future examination.

Control of the cell cycle, particularly G1-to-S progression, is reliant upon ROS signaling [30,31,47,66]. Ligand-growth factor receptor interactions initiate a transient rise in ROS, specifically H_2_O_2_, which acts as a second messenger to potentiate downstream signaling along pathways such as the mitogenic RAS-RAF-MEK-ERK cascade [67]. Mitochondria generate low amounts of ROS off the electron transport chain [32] and mitochondrial ROS is important to promoting cell proliferation [31,33,34,35]. Of relevance, tumor cells, particularly in highly aggressive cancers, demonstrate hybrid metabolism by using glycolysis and OXPHOS to meet the energetic demands of sustained growth and cell turnover [51,68]. Upon MTLN KD, we observed a decline in steady-state H_2_O_2_ levels. This suggests that under control conditions, MTLN contributes to generating the ROS needed for ERK activation. Several reports indicate that MTLN promotes OXPHOS [7,10,11,15]. Thus, lowered H_2_O_2_ levels could be due to a metabolic shift away from OXPHOS after MTLN loss. Since A549 cells harbor a hyperactive KRAS mutation, the shutdown of p-ERK upon MTLN KD is likely regulated at the level of ERK phosphorylation/dephosphorylation. Phosphatases are known to undergo reversible regulation by H_2_O_2_-mediated oxidation of cysteine thiol groups to inhibit their activity [48]. This includes DUSP6, the dominant ERK-specific phosphatase [69]. In addition, ERK itself is sensitive to H_2_O_2_-mediated oxidation [70,71]. A recent study shows that both MEK and ERK shift localization to mitochondria to facilitate ERK phosphorylation and subsequent translocation to the nucleus [72]. That study also showed that low-dose H_2_O_2_ induces the oxidation of ERK at two specific cysteine residues for enhanced binding to p-MEK and resultant ERK activation [72]. MTLN could assist in delivering ROS to the cytosol and/or could expedite local oxidation of mitochondria-docked ERK (Figure 5). Continued investigation is warranted to delineate these events.

Along with the decline in p-ERK, we detected a fall in MYC and rise in p27^Kip1^ protein levels in response to MTLN KD. MYC is a transcription factor that operates in parallel with RAS-RAF-MEK-ERK to upregulate cyclin and CDK gene expression and to downregulate genes that oppose cell cycle progression [25,73]. p27^Kip1^ antagonizes cell G1/S transition by inhibiting CDK activity in the nucleus [23,74]. The nuclear localization and abundance of p27^Kip1^ are tightly regulated by post-translational modifications [24,75,76,77,78]. With MTLN KD, diminished p-ERK, CDK2, and MYC could jointly explain the elevated p27^Kip1^, as these proteins contribute to repressing p27^Kip1^ transcription and/or promoting p27^Kip1^ nuclear export and destabilization through phosphorylation and proteasomal degradation [24].

The one outcome of MTLN KD that contradicts the prototypical G1-arrested profile [34,47,74] is the elevation in cyclin D1. ERK activation supports cyclin D1 transcription [79,80] and is dampened after MTLN KD. This suggests that the rise in cyclin D1 occurs by a post-transcriptional mechanism, such as mRNA and/or protein stabilization. Conceivably, this could involve a weakened expression of the miR-17-92 oncogenic cluster which is integral to regulatory networks in cancer [81,82]. Both cyclin D1 [83] and MTLN [14] are have been validated as miR-17 targets. Moreover, work from our lab has further established MTLN as a functional miR-17 target; we previously observed Ago2 binding at the portion of MTLN mRNA encompassing the miR-17 target site [84]. In follow-up experiments, we found that the addition of miR-17 mimic strikingly diminished the MTLN protein levels in the A549 cells (Appendix A). MYC and E2F1 transcriptionally activate the miR-17-92 cluster in a feedback loop to negatively regulate E2F1 translation [85]. The drop in MYC level and presumed decline in miR-17 upon MTLN KD could explain the cyclin D1 elevation. An interesting possibility is that MTLN participates in this loop in an attempt to maintain its own levels within an optimal range. Notably, cyclin D1 has non-canonical functions outside of the nucleus [86]. For example, cytoplasmic membrane-associated cyclin D1 fosters invasiveness and metastasis in cancers [87,88], while mitochondria-localized cyclin D1 alters glucose metabolism [89,90]. We envision possible direct as well as indirect MTLN-cyclin D1 interplays and are enthusiastic to explore these prospects.

Lastly, when the A549 cells were exposed to inhibitors of hydroperoxide scavenging (BSO + AF), these drugs were less toxic to the MTLN KD cultures compared to the controls. We propose several factors possibly contributing to this outcome. AF primarily creates a ROS surge by blocking the breakdown of mitochondria-derived H_2_O_2_ [91]. During the 3 h AF exposure, metabolically quiescent cells with lowest mitochondrial H_2_O_2_ might experience a sub-lethal rise in ROS. This is consistent with the identification of a subpopulation of cells that showed only faint staining with the CellROX Orange sensor. This population was more abundant in the siMTLN cultures (determined at the 5 μM AF dose). Based on CellROX staining, ROS was significantly elevated at both 2.5 and 5 μM AF in the siMTLN cultures but only at 5 μM for the siCont cultures. The low/negative signal across the siCont monolayer at 2.5 μM was difficult to reconcile given the poor survival (<10% surviving fraction) in the colony forming assay with this same drug treatment. Perhaps the oxidation of the CellROX sensor for 30 min of the 3 h AF treatment does not adequately reflect cumulative ROS. Also, the ROS species detected by CellROX Orange may not correspond to the most toxic forms of ROS [92]. Future time-resolved evaluation using a more sensitive and dynamic H_2_O_2_ indicator [93] should better parse out events during AF exposure. Both the siCont and siMTLN groups sustained substantial DNA damage after the BSO + AF treatment, based on elevated pS139-H2AX. An additional factor that could contribute to the improved survival of MTLN-silenced cells is DNA repair. Cyclin D1, augmented by MTLN silencing, has been reported to assist in the recruitment of DNA repair complexes to sites of damage [63]. Given this prospect, the simultaneous silencing of MTLN and cyclin D1, with or without additional chemotherapeutic drug exposure, may prove to more fully eradicate NSCLC survival and growth.

Collectively, our findings reveal a role for MTLN in perpetuating NSCLC cell proliferation, attributed in part to enhancing H_2_O_2_ output. Our findings do not rule out additional pathways through which MTLN may contribute to cancer cell biology. MTLN loss has been associated with changes in mitochondrial calcium [7], ER contacts [18], mitochondrial superoxide [7,18], fatty acid oxidation [9,12,19], retrograde JUN signaling [65], and lipid metabolism [10,16,19], all with the potential to influence cancer progression or sensitivity to anti-cancer drugs [94,95,96,97,98]. Furthermore, it would be of interest to evaluate MTLN gain-and-loss effects on glucose metabolism and the potency of antioxidant pathways. Cancer cells shunt glucose into the pentose phosphate pathway for NADPH synthesis to counter oxidative stress [99] and, purportedly, as a step towards conversion to a cancer stem cell state [100]. In sum, cancer cells evolve through complex interactive pathways to support cycle progression and survival, and it is becoming increasingly apparent that mitochondria can serve as vital hubs along these pathways. The continued discovery of novel players and vulnerable components, such as MTLN, has the potential to lead to the development of effective multi-hit approaches for cancer therapy.

## 4. Materials and Methods

### 4.1. Online Bioinformatic Data Acquisition and Analyses

The MTLN gene expression box plot across cancerous and noncancerous tissues was generated from the TNMplot website, using their online pan-cancer gene expression analysis tool designed to compare normal and tumorous samples across 22 tissue types [39]. Transcript expression data utilized by TNMplot are acquired from the Gene Expression Omnibus of the National Center for Biotechnology Information (NCBI-GEO), The Cancer Genome Atlas (TCGA), Therapeutically Applicable Research to Generate Effective Treatments (TARGET), and The Genotype–Tissue Expression (GTEx) repositories [39]. Statistical comparison of non-cancerous versus cancerous tissues was performed by the Mann–Whitney U test (auto-generated by TNMplot). Font style and color scheme alterations were applied to the online-generated plot for presentation, as shown in Figure 1A.

The data used to generate the graph for MTLN gene expression (LUAD compared to normal lung tissue) were generated from the OncoDB website [40] using their Gene Expression Profile tool. The normalized TPM values (540 LUAD and 59 normal lung samples) were downloaded from OncoDB, and GraphPad Prism software 10.4.1 was used to generate the Figure 1B scatter plot and for statistical analysis (unpaired *t*-test).

The Kaplan–Meier plot for LUAD cancer survival was generated from the Kaplan–Meier Plotter website using their KM plotter tool [41,42]. The following parameters were selected for Lung Cancer analysis: LINC00116 228614_at Affymetrix probe; auto-select best cutoff; overall survival (OS); subtype restriction LUAD; no restriction on treatment groups; no restriction on database selection; and exclusion of biased arrays. Of note, we did not apply restrictions/filters for tumor grade/stage or treatment (chemotherapy or radiation) for LUAD KM plot analyses since post-filter sample numbers were too low for meaningful analyses. Statistics were automatically generated by the KM plotter tool; cox regression multivariate analysis was performed with a cut-off value of 242, probe expression range of 13–2817, and 1% FDR. Some font style and color scheme alternations were applied to the online-generated graph for presentation shown in Figure 1C.

### 4.2. Cell Culture and Knockdown

Human A549 (ATCC, CCL-185, Manassas, VA, USA) and H1299 (ATCC, CRL-5803) cells were maintained in Gibco DMEM/F-12 (cat#11320033,Thermo Fisher Scientific [TFS], Waltham, MA, USA) with 100 U/mL penicillin, 0.1 mg/mL streptomycin and 10% FBS in a cell culture incubator in standard conditions (humidified, 37 °C with 5% CO_2_). Cells were kept at a low passage and regularly passaged prior to 100% confluency. For MTLN silencing, a dicer-ready 27-mer RNA oligo pair targeting the human *MTLN* open reading frame (siMTLN) was purchased from Integrated DNA Technology (IDT; Coralville, IA, USA), with the following sequences:

5′rArCrUrGrCrArGrUrUrGrUrCrCrGrUrGrCrUrArGrUrArGCC3′

5′rGrGrCrUrArCrUrArGrCrArCrGrGrArCrArArCrUrGrCrArGrUrGrU3′.

Cells were transfected with 40 nM siMTLN or negative control (siCont) (IDT cat# 51-01-14-03) dsiRNA oligos using Lipofectamine 2000 (TFS), according to the product instructions, or with 12.5 nM oligos using RNAiMAX (TFS), according to product instructions, and Opti-MEM media (TFS) was used for complex formation.

### 4.3. Immunofluorescent Staining

A549 cells were cultured and transfected as described above. At 48 h post-transfection, the cells were incubated in culture media containing 200 nM MitoTracker Red CMXRos (TFS) for 20 min, washed one time with media, then fixed by incubation in 4% PFA in PBS for 10 min at 37 °C. They were then washed in PBS and blocked and permeabilized by incubation for 1 h with PBS containing 10% normal goat serum (NGS) and 0.1% triton. The blocking buffer was removed and the cells were incubated overnight at 4 °C with affinity purified custom rabbit antibody against MTLN c-terminus [7], diluted 1/300 in PBS containing 1% NGS and 0.1% triton. The cells were washed in PBS, stained with Alexa 488-conjugated goat anti-rabbit antibody at 1/2000 for 1 h at RT, then washed again with PBS. Fluorescent images were captured using an Olympus IX70 microscope with a 60× objective equipped with an Olympus DP70 digital camera (Olympus Corporation of Americas, Center Valley, PA USA). Channel brightness/contrast adjustments were applied equally across groups using ImageJ (Fiji2) software 2.3.1. Pearson’s coefficient for co-localization was determined using the ImageJ JACoP plug-in with Costes’ automatic threshold.

### 4.4. Cell Growth Assay

A549 cells were seeded at 30,000 cells per well into 12-well culture dishes. The following day (day 1), the cells were transfected with siRNA oligos using RNAiMax as described above. On the indicated day, cells were washed twice with PBS, lifted with trypsin, and counted on a Beckman Z1 Coulter Particle Counter (Beckman Coulter Inc., Brea, CA USA).

### 4.5. Cell Cycle Analysis by DNA Content

A549 cells were plated in 60 mm dishes at 300,000 cells per dish to limit contact inhibition at the time of collection. The following day, the cells were transfected with siRNA oligos (siCont or siMTLN) using RNAimax as described above. At 48 h post-transfection, the cells were lifted with trypsin and pelleted by centrifugation. The cells were resuspended in 100 μL PBS and then fixed by the dropwise addition of 3 mL of −20 °C 70% EtOH, followed by an incubation for 1 h at 4 °C. After fixation, the cells were washed thrice with PBS and suspended in 100 μL PBS with 50 μg RNase A, followed by the addition of 10 μg propidium iodide in 100 μL PBS and a 30 min incubation at 4 °C. DNA content analysis was determined by flow cytometry at the University of Iowa Flow Cytometry Facility using a Becton Dickinson LSR II (BD Biosciences, Franklin Lakes, NJ, USA) and ModFit software version 4.0.

### 4.6. Western Blot

Lysates were collected in RIPA buffer (cat# R0278, Sigma-Aldrich, St. Louis, MO, USA) or LDS lysis buffer (1% lithium dodecyl sulfate in 50 mM Tris, pH 7.5) with freshly added Roche Complete Protease Inhibitors and PhosSTOP phosphatase inhibitors (Roche, Basel, Switzerland). Protein concentrations were determined by a BCA assay (Pierce BCA Protein Assay Kit, REF #23227, TFS) and the concentrations were equilibrated. NuPAGE sample buffer and reducing agent (Invitrogen) were added to 1×, and the samples were heated at 70 °C for 10 min and resolved on NuPAGE 4–12% Bis-Tris gels using MES or MOPS running buffer (TFS). Proteins were transferred to 0.2 μm PVDF or nitrocellulose membranes using NuPAGE transfer buffer with 10% ethanol. The membrane was blocked with 5% milk powder in TBS with 0.1% Tween-20 (TBST) and incubated with primary antibody overnight at 4 °C with rocking. Membranes were washed with TBST, incubated for 1 h at room temperature with the HRP-conjugated secondary antibody, washed, and developed with Radiance Plus chemiluminescence HRP substrate (Azure Biosciences, Dublin, CA, USA); the images were collected using an iBright 1500 system (TFS). For some experiments, the membrane was stained for total protein with Ponceau S (5% *v*/*v* glacial acetic acid and 0.1% *w*/*v* Ponceau S) immediately after transfer, imaged on the iBright system, and then destained with 0.1 M NaOH before proceeding as described above with the blocking and antibody staining steps. After blotting for the phosphorylated proteins, the membranes were stripped using Restore (TFS), re-blocked, and then immuno-blotted for the total protein.

Antibodies used for Western blot were: anti-β-actin (Sigma-Aldrich, A5411 ascites, 1/10,000); anti-p27^Kip1^ (BD Biosciences, #610241 clone 57, 1/5000); anti-cyclin D1 (Cell Signaling Technologies [CST] #2922, 1/500, Danvers, MA, USA); anti-phospho-H2AX S139 (CST #2577, 1/1000); anti-H2AX (R&D Systems, MAB3406, 0.5 μg/mL, Minneapolis, MN, USA); anti-phospho-AKT T308 (CST #2965, 1/1000); anti-AKT (CST #9272, 1/1000); anti-ROMO1 (Origene, TA505580, clone OTI2C12, 0.5 μg/mL, Rockville, MD, USA); anti-p21 Waf1/Cip1 (CST #2947, 1/1000); anti-ERK (CST #9102; p44/42 MAPK [Erk1/2]); anti-phospho-ERK T202/Y204 (CST #9101; p44/42 MAPK [Erk1/2]); anti-phospho-RB S807/811 (CST #9308, 1/1000); anti-RB (CST #9309, clone 4H1, 1/2000); anti-CDK2 (Abcam ab6538, Cambridge, UK); anti-cyclin E1 (CST #20808, clone D7T3U, 1/1000); anti-MYC (CST #5605, clone D84C12, 1/1000); anti-MTLN (custom rabbit antibody against MTLN c-terminus [7], 1/2000); anti-beta-tubulin (DSHB, clone E7, 0.5 μg/mL, Iowa City, IA, USA); and anti-GAPDH (Abclonal #AC002, 1/10,000, Woburn, MA, USA).

### 4.7. H_2_O_2_ Measurement with roGFP2-Orp1

Adβal (Ad5CMVcytoLacZ; VVC-U of Iowa-3554) was purchased as an in-stock vector from the University of Iowa Viral Vector Core (VVC; Iowa City, IA, USA). Cytosol- and mitochondria-localized roGFP2-Orp1 H_2_O_2_ sensors were subcloned from plasmids donated to Addgene from Tobias Dick [56] (Addgene plasmids #64991 and #64992, Watertown, MA, USA) into a VVC adenovirus shuttle vector (VVC G0688; pacAd5CMVmcspA) using standard cloning techniques, and vector particles (Adcyto-roGFP2-Orp1 and Admito-roGFP2-Orp1) were generated by the VVC. The A549 cells were seeded at 25,000 cells per well in 48-well culture dishes and transfected the next day with control (siCont) or MTLN-directed (siMTLN) RNAi oligos using RNAiMAX as described above. The transfection reagent was removed after 5 h and replaced with culture media containing Adcyto-roGFP2-Orp1, Adcyto-roGFP2-Orp1, or Adβgal at a multiplicity of infection of 10–15 PFU per cell. Two days later, the culture media was exchanged for imaging media (HBSS with calcium and magnesium, supplemented with 30 mM HEPES, 2% FBS, and 1 mM pyruvate). Fluorescence measurements were then collected for each culture well on a CLARIOstar Plus plate reader (BMG Labtech, Ortenberg, Germany) at ex/em 400/520 and ex/em 485/520. The background fluorescence (mean reading from Adβgal-transduced wells of siCont and siMTLN groups) was subtracted from the Adcyto-roGFP2-Orp1 and Adcyto-roGFP2-Orp1 readings, and the ratio of ex/em 400/520 to ex/em 485/520 (oxidized/reduced) was calculated for each well as an indicator of the relative roGFP2 oxidation state.

### 4.8. H_2_O_2_ Dose Response

For this assay, A549 cells were cultured in DMEM media (Cat#11965092, TFS) supplemented with 100 U/mL penicillin, 0.1 mg/mL streptomycin, and 10% FBS. The A549 cells were seeded at 2.5 million cells into 100 mm dishes and transfected the following morning with siCont or siMTLN oligos using RNAiMAX as described above. After 6 h, the cells were lifted with trypsin, counted in a Coulter Counter, and then re-seeded at 50,000 cells per well into 24-well culture dishes. The following day, H_2_O_2_ was added to the wells to produce different final concentrations (0, 7.5, 15, 30, 60, 120, 240, and 480 μM). After 3 h, the cells were washed with culture media, then cultured in fresh media for a further 24 h, lifted with 0.25% trypsin, and diluted in media before determining the total yields using a Coulter Counter.

### 4.9. BSO + AF Treatment and Colony Formation Assay

The A549 cells were seeded in 6-well plates at 100,000 cells per well and transfected the next day with siCont or siMTLN oligos at 40 nM using Lipofectamine 2000 as described above. After 4 h, the transfection media was replaced with culture media. At 48 h post-transfection, BSO was added to a final 100 μM concentration. The following day, AF was added to a final concentration of 2.5 μM. At 3 h post-AF exposure, the cells were lifted into 0.25% trypsin and diluted in media, and the cell concentrations were then determined using a Coulter Counter. The cells were then plated into 6-well dishes in complete culture media in triplicate wells at densities ranging from 150 to 100,000 cells per dish. After 10 days, the cells were fixed with 70% ethanol and stained with Coomassie blue solution (0.4 *w*/*v* in 1:4:1 methanol, water, acetic acid). Colonies were considered as more than 50 cells and were counted to calculate the plating efficiency (for non-drug treated) and the surviving fraction (for BSO + AF treated) as follows:

Plating Efficiency = number of colonies counted ÷ number of cells plated

Plating Efficiency (%) = number of colonies counted ÷ number of cells plated × 100

Surviving Fraction, normalized = number of colonies counted ÷ number of cells plated ÷ Plating Efficiency

### 4.10. BSO + AF Treatment and LDH Release

The A549 cells were seeded at 25,000 cells per well in 48-well culture dishes and transfected the next day with control (siCont) or MTLN-directed (siMTLN) RNAi oligos using RNAiMAX as described above. At 48 h post-transfection, the media was changed to media with or without 100 μM BSO. The following day, cultures containing BSO were treated for 3 h with 2.5 μM AF. Some wells were treated with 2 mM N-acetyl-L-cysteine (NAC); NAC was added 30 min prior to AF addition and maintained during the 3 h AF treatment. After 3 h, conditioned media was collected for an LDH assay (CytoScan LDH Cytotoxicity Assay, S/N: 10089706, G-Biosciences, St. Louis, MO, USA). For the determination of maximal LDH release, the non-treated siCont and siMTLN cells were lysed by the addition of Triton X-100 to 1%, and the lysates were clarified by centrifugation for use in the LDH assay. For LDH determination, the conditioned media and lysate samples, as well as the “media alone” background controls, were diluted and tested in triplicate in the colorimetric LDH activity assay according to the kit instructions. OD readings from the conditioned media and lysates were background-corrected by subtracting the OD reading obtained from the culture media alone. Using background-corrected OD readings, LDH release was determined as the ratio of conditioned media to lysate for the appropriate group (siCont or siMTLN).

### 4.11. CellROX Orange Analysis

A549 cells were seeded at 25,000 cells per well into 24-well dishes and transfected the next day with siCont or siMTLN oligos at 40 nM using Lipofectamine 2000. The transfection media was replaced with culture media after 4 h. At 48 h post-transfection, the media was changed to media with or without BSO at 100 μM. At 24 h post-BSO addition, AF was added to the BSO-containing wells to final 2.5 or 5.0 μM. At 2.5 h post-AF, probenecid (water soluble, TFS) was added (to 2.5 mM) to all the cultures, followed by CellROX Orange (to 4 μM), and the cultures were incubated for a further 30 min. The wells were washed twice with imaging buffer (HBSS with calcium and magnesium, supplemented with 30 mM HEPES, 2% FBS and 2.5 mM probenecid), then 500 μL imaging buffer was added per well for live cell imaging using a Nikon Eclipse Ti2 inverted microscope with LED illumination (Nikon Instruments Inc., Melville, NY, USA) and NIS Elements software version 6.10.01. Three images per well (technical replicates) were captured using the 10× objective at identical exposure settings. Fluorescence intensities for each image were determined using ImageJ Fiji2. Total cell counts and negative/dim cells were tallied using ImageJ Fiji2. Single values used for statistical analyses (*n* = 3 per group) were obtained by averaging the three technical replicates. Image brightness was adjusted equally across images using Adobe Photoshop 26.3 software for Appendix A.

### 4.12. Treatment with microRNA Mimic

A549 cells were seeded at 60,000 cells per well into 24-well dishes. The following day, cells were transfected with mirVana miRNA mimics (TFS) for miR-17 (hsa-miR-17-5-p) or negative control #1 using RNAiMAX, according to the product instructions, for final miRNA mimic concentrations of 10 nM or 50 nM. At 48 h post-transfection, cells were harvested into RIPA lysis buffer supplemented with protease and phosphatase inhibitors and clarified by centrifugation, and the protein concentrations were then determined by BCA assay. Concentrations were equilibrated prior to performing a Western blot as described above.

## Figures and Tables

**Figure 1 ijms-26-01939-f001:**
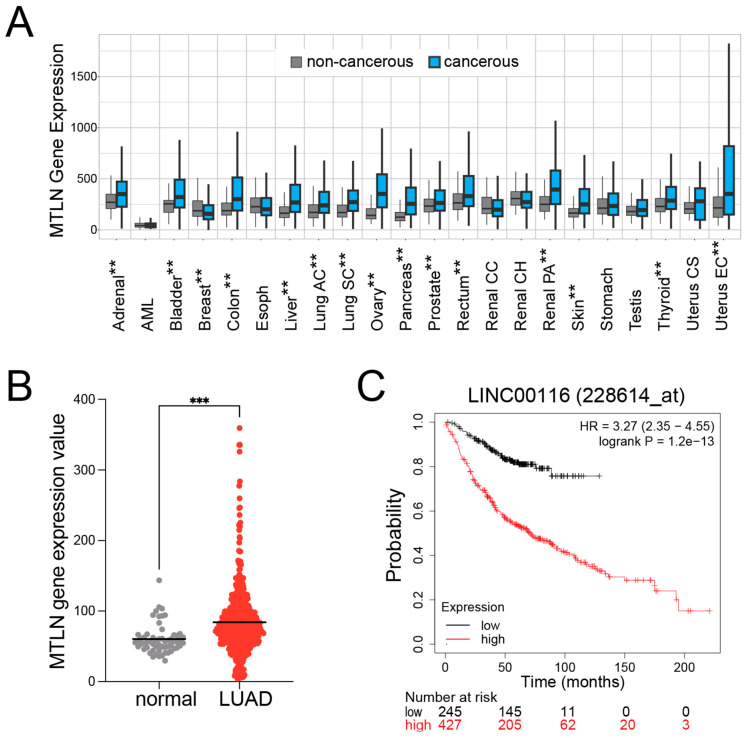
MTLN gene expression is increased in most cancer types and inversely correlates with overall survival in lung adenocarcinoma (LUAD). (**A**) MTLN transcript levels are elevated in most cancers. Profile plot of MTLN gene expression (normalized, using TPM) across 22 cancerous and corresponding non-cancerous tissue types was generated with the TNMplot website tool. Significant differences determined by Mann–Whitney U test (** *p* < 0.01). (**B**) MTLN gene expression is elevated in LUAD compared to normal lung tissue. Data derived from the OncoDB website using the Gene Expression Profile tool. (*n* = 540 for LUAD and *n* = 59 for normal lung tissue; normalized, using TPM) *** *p* < 0.001 by unpaired *t*-test. (**C**) High MTLN gene expression predicts poor survival in LUAD. Kaplan–Meier Plot was generated using the KM-plotter tool on the Kaplan–Meier Plotter website; cox regression statistical analysis applied with 1% FDR (*p* = 1.2 × 10^−13^).

**Figure 2 ijms-26-01939-f002:**
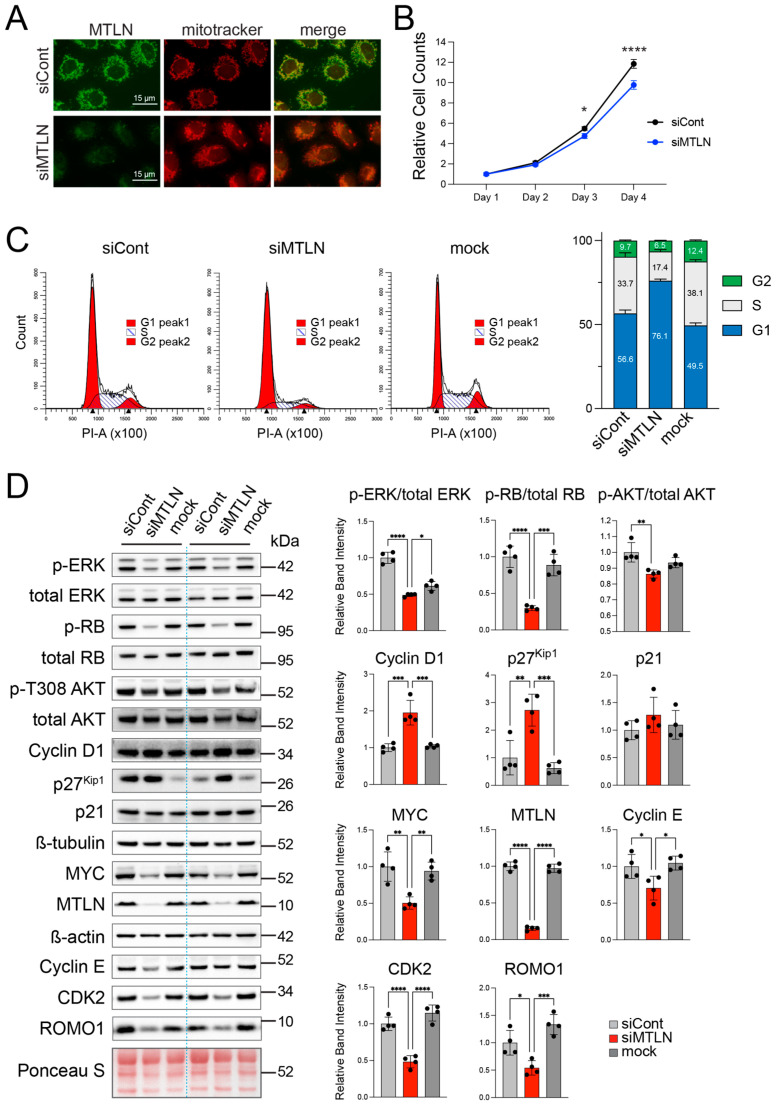
MTLN knockdown induces G1 phase cell cycle arrest in A549 cells. (**A**) MTLN localizes to mitochondria in A549 cells. At 48 h post-MTLN KD (siMTLN) or control treatment (siCont), A549 cells were incubated with Mitotracker (red) then fixed and stained for MTLN (green) using a custom antibody. (**B**) MTLN KD slows proliferation rate. Cultures were treated with siMTLN or siCont the day after seeding (=day 1) then harvested and counted at different time points. Bars show mean ± standard deviation (SD) (*n* = 4 per group at each time-point) * *p* < 0.05 and **** *p* < 0.0001 for siCont compared to siMTLN by 2-way ANOVA with multiple comparisons. (**C**) PI staining shows a block in G1/S transition upon MTLN silencing. A549 cells were collected at 48 h post siMTLN, siCont or mock treatment and nuclei stained with propidium iodide (PI) and analyzed for DNA content by flow cytometry. Representative flow cytometry histograms are shown on the left; distribution of cells in G1/S/G2 for each group (mean ± SD) are shown on the right (*n* = 3 siCont; *n* = 3 siMTLN; *n* = 2 mock). (**D**) Cycle-related protein changes reflect G1 arrest. Cell lysates harvested at 48 h after siMTLN, siCont, or mock treatment were assayed for various proteins by Western blotting. Bars are mean ± SD band intensities relative to siCont. Phosphorylated band intensities were normalized to total band intensity of the same protein within the same lane. Other protein bands were normalized to loading control within the same lane (β-tubulin for cyclin D1, p27^Kip1^ and p21; β-actin for MYC and MTLN; Ponceau S for cyclin E, CDK2 and ROMO1). (*n* = 4 per group) * *p* < 0.05, ** *p* < 0.01, *** *p* < 0.001 or **** *p* < 0.0001 by 1-way ANOVA with multiple comparisons.

**Figure 3 ijms-26-01939-f003:**
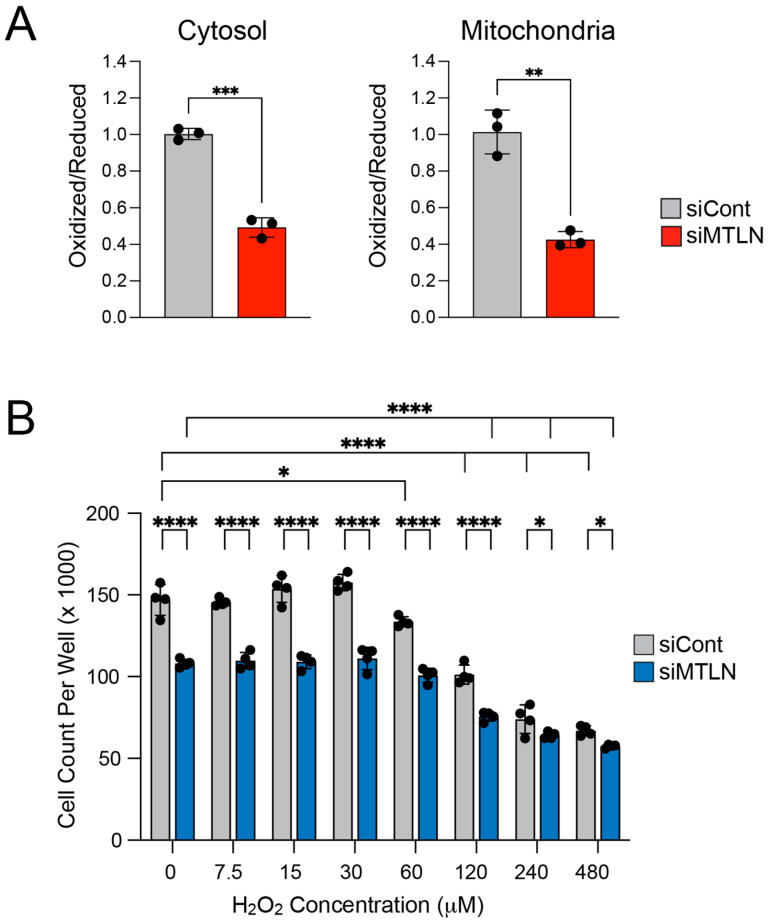
MTLN silencing decreases endogenous H_2_O_2_ levels and reduces sensitivity to exogenous H_2_O_2_ toxicity. (**A**) The cytosol- and mitochondria-localized roGFP2-Orp1 H_2_O_2_ sensors are less oxidized in siMTLN compared to siCont cultures. Bars show mean ± SD (*n* = 3) ** *p* < 0.01, *** *p* < 0.001 by unpaired *t*-test. (**B**) Distinct dose–response effects of H_2_O_2_ pulse on cell proliferation in siMTLN compared to siCont cultures. Cultures were exposed to the indicated amount of H_2_O_2_ for 3 h and then harvested and counted 24 h later. Bars show mean ± SD (*n* = 4) * *p* < 0.05, **** *p* < 0.0001 by 2-way ANOVA with multiple comparisons.

**Figure 4 ijms-26-01939-f004:**
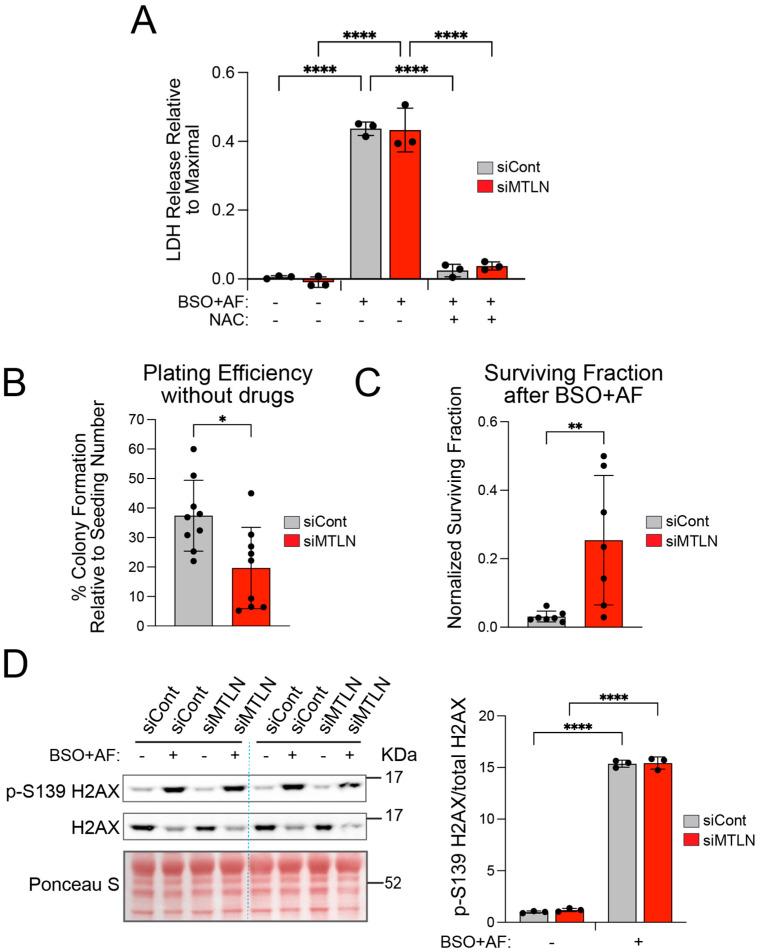
MTLN knockdown reduces efficacy of BSO + AF chemotherapy. (**A**) Extent of LDH leak at 3 h post-AF (2.5 μM) addition is similar for siCont and siMTLN cultures. At 48 h post siMTLN or -siCont, BSO (100 μM) was added, then AF (2.5 μM) was added the following day. At 3 h post-AF addition, lactate dehydrogenase (LDH) leak to the media was measured as an indicator of cell death. Data are expressed relative to maximal possible release. N-acetyl cysteine (NAC) antioxidant was added 30 min prior to AF addition. Bars show mean ± SD (*n* = 3) **** *p* < 0.0001 by 1-way ANOVA with multiple comparisons. (**B**) MTLN silencing impairs plating efficiency. At 3 days after siCont or siMTLN treatment, A549 cells were lifted with trypsin and reseeded at low density to determine colony forming capacity (CFC). Bars show % plating efficiency mean ± SD (*n* = 9) * *p* < 0.05 by unpaired *t*-test. (**C**) MTLN silencing provides partial resistance to BSO + AF toxicity. Cultures were treated with siRNA and BSO + AF, as described in (**A**), then lifted with trypsin and reseeded to determine CFC. Bars show mean ± SD surviving fraction normalized to plating efficiency for each group (*n* = 7) ** *p* < 0.01 by unpaired *t*-test. (**D**) BSO + AF exposure elicits similar DNA damage in siMTLN and siCont cultures. Lysates harvested at 3 h post-AF addition were assayed by Western blotting for p-S139 H2AX as a marker of DNA damage. Bars show mean ± SD of p-S139 H2AX/total H2AX relative band intensities (*n* = 3 per group) **** *p* < 0.0001 by 1-way ANOVA with multiple comparison.

**Figure 5 ijms-26-01939-f005:**
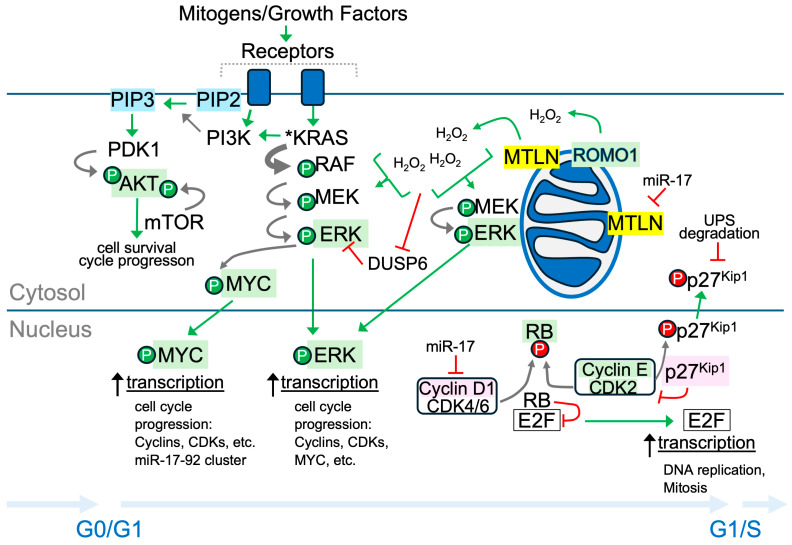
Working model of MTLN contribution to G1/S transition. Schematic shows a simplified diagram of signaling pathways that regulate cell cycle progression through G1 in KRAS mutated LUAD cancer cells. Our model suggests that MTLN promotes mitochondrial H_2_O_2_ release to support ERK activation and subsequent downstream signaling events. Green arrows signify steps that promote cycle progression. Red flathead lines signify inhibition of target protein activity or abundance. Gray arrows show phosphorylation steps; thick gray arrow represents sustained kinase activity of mutated KRAS (*KRAS). Encircled “P” indicates phosphorylation: green is activating; red is inactivating/destabilizing. Signaling phospholipids are highlighted blue. The protein central to this study (MTLN) is highlighted yellow. Green highlighted proteins were decreased upon MTLN KD (p-ERK, MYC, p-AKT, p-RB, cyclin E, CDK2, ROMO1). Pink highlighted proteins were elevated upon MTLN KD (p27^Kip1^, cyclin D1). Non-highlighted were not examined. Up black arrow signifies increase.

## Data Availability

The original contributions presented in this study are included in the article/Appendix A. Further inquiries can be directed to the corresponding authors.

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
