# Peer review of "Mitoregulin Promotes Cell Cycle Progression in Non-Small Cell Lung Cancer Cells"

_ijms, 2025, doi:10.3390/ijms26051939_

Round 1

Reviewer 1 Report

Comments and Suggestions for Authors

1.In Section 2.1, for the high expression of MTLN in lung cancer, the authors could try to use western blot test or detect mRNA expression level, and use normal cells HL-7702 or WI-38 as control, which is more intuitive and more beneficial for readers to understand.

2.As for Fig.2A, how well do MTLN and mitotracker fit? The author should represent it in a bar chart.

3.MTLN can accumulate in mitochondria, so what is the effect of siMTLN, siCont or mock on mitochondrial membrane potential?

4.As for Section 2.4, the effects of DNA damage are most intuitive using the comet assay.

5. The sequences such as siMTLN, siCont, etc. should be listed in the lab section or in the support information file, which is more reader-friendly.

Reviewer 2 Report

Comments and Suggestions for Authors

This is very interesting article and presented results are very interesting.

The section 4 Materials and Methods must be improved. The databases and the used methods are well described, but there are no data about the patients which tissues have been used, i.e. stage of disease, the tissues used in the analysis were obtained before the treatment or after the treatment was introduced. I suggest that the authors add a paragraph about the patient’s characteristics. If the samples were obtained after the treatment was introduced, which treatment was applied? Did it have impact on results? What about driver mutations which can be found in patients with lung adenocarcinoma (EGFR, ALK, ROS1…)? It is well known that patients with driver mutations have better outcomes. If the authors included patients with driver mutations, did it have impact on results? What about stage of disease, did it have impact on results? Patients with different stage of disease have different outcomes. 

Reviewer 3 Report

Comments and Suggestions for Authors

Overall very nice work and nicely executed as well as drafted. My specific comments are: 

1. Cell division is a universal phenomenon for almost all cells at some stage of their life. This illustrates that mitoregulin is crucially involved with cell cycle regulation for normal and cancerous cells. The manuscript title "Mitoregulin Promotes Cell Cycle Progression in Non-Small Cell Lung Cancer Cells" seems to a role in NSCLC cells only, which is not the case as shown in the manuscript data. The author may rephrase the title accordingly to justify the statement. 

2. Besides cancerous cells or cancer patients, in what other conditions has mitoregulin been reported to be elevated? If applicable, how does the author interpret this elevation of mitoregulin and cancerous cells? Like which type of confirmatory test will ensure their exclusive role for cancerous cells or NSCLC. 

Round 2

Reviewer 2 Report

Comments and Suggestions for Authors

No further comments

Author Response

Thank you for the review of our revised manuscript. We appreciate your contribution.